

# Attitudes and intentions of parents towards the COVID-19 vaccine for their children at a special moment of the pandemic

Nurcan Çelik Odabaşı[1,*], Ali Tayhan[2,*] and Gulengul Mermer[3]

[1] Department of Midwifery/Faculty of Health Sciences, Celal Bayar University, Manisa, Turkey
[2] Department of Public Health Nursing/Faculty of Health Sciences, Celal Bayar University, Manisa, Turkey
[3] Department of Public Health Nursing/Faculty of Nursing, Ege University, Izmir, Turkey
[*] These authors contributed equally to this work.

## ABSTRACT

**Background**. This research was conducted following the FDA's approval of the COVID-19 vaccine for children aged 5 to 11. Our study aimed to evaluate parents' intentions regarding vaccinating their children in response to the pandemic situation. This period was crucial for understanding parents' initial reactions to health authority recommendations.

**Methods**. This descriptive cross-sectional study was conducted with 940 parents. The data were obtained using the Personal Information Form, the COVID-19 Anxiety Scale, and the Scale of Attitudes Towards COVID-19 Vaccine. Descriptive data analysis involved calculating frequency, percentage, mean, and standard deviation values. We employed Student's t-test, one-way ANOVA tests for analysis. Additionally, various characteristics were analyzed using the two-way logistic regression method.

**Results**. In the study, 85.1% of the parents reported having received the COVID-19 vaccine themselves. However, only 20% stated they would accept the COVID-19 vaccine for their children. The intention to vaccinate their children was influenced by factors such as the parent's age, level of COVID-19 anxiety, vaccination status, and knowledge about the COVID-19 vaccine.

**Conclusion**. Despite the FDA's declaration, it was determined that more than two-thirds of parents were hesitant about the COVID-19 vaccine for their children.

## INTRODUCTION

COVID-19, a global pandemic that emerged in 2019, precipitated a significant worldwide public health crisis (*Ellithorpe et al., 2022*). According to the World Health Organization (WHO), as of January 2023, the disease had resulted in 6.7 million deaths worldwide (*World Health Organization, 2023*). The first case was seen in Turkey on March 10, 2020, and as of March 2023, 102,174 people lost their lives due to COVID-19 (*Varol & Tokuç, 2020*). As a result of the understanding of the severity of the disease, research was immediately initiated to develop vaccines and improve herd immunity (*Sharma et al., 2020*). Some

Corresponding author
Ali Tayhan, ali.tayhan@cbu.edu.tr

vaccines resulting from the initial vaccine studies were recommended by the WHO and subsequently implemented in many countries. During this period, 85% of adults in Turkey were given free COVID-19 vaccines (*Republic of Turkey Ministry of Health, 2023*). The WHO recommended that elderly people be given priority in the vaccination program because the disease is more severe in older individuals. In December 2021, the US Food and Drug Administration (FDA) recommended that children, like adults, be vaccinated against COVID-19 (*COVID-19 Vaccine for Emergency Use in Children, 2021*). Following these recommendations, the vaccination program against COVID-19 for children began to be implemented in Italy on January 17, 2022. However, less than 40% of children aged 5–11 were vaccinated against COVID-19 (*Sacco et al., 2022*). A study conducted in the USA before the 2021 FDA declaration found that less than half of parents planned to vaccinate their children against COVID-19 (*Szilagyi et al., 2021*). A study conducted in turkey in the early moments of the pandemic found that 38% of parents planned to have their children vaccinated if the COVID-19 vaccination was recommended for their children (*Yılmaz & Sahin, 2021*). When examining the Italian example, it is noteworthy that while the acceptance rate of childhood vaccines in Italy was 93% (*Sabbatucci et al., 2021*), the low vaccination rates among children aged 5–11 against COVID-19 underscored a situation of vaccine hesitancy specifically for the COVID-19 vaccine. Vaccine hesitancy can have different causes and is known to be influenced by different contexts. *Williams (2014)* emphasizes that, in general, people's anxiety about the consequences of the disease and their attitudes towards the vaccine were effective in vaccine decisions. It is known that disease anxiety often positively affects preventive health practices such as vaccination. In addition, various studies indicated that parents' anxiety and their attitudes towards vaccination also affected their decisions about the vaccine recommended for their children (*McNeil & Purdon, 2022*). On the other hand, *Opel et al. (2023)* stated that with the COVID-19 pandemic, the feeling of suspicion towards other health practices, especially vaccination, increased in the society, and that the pandemic may thus leave a bad legacy on future health practices.

When the behaviors and decision mechanisms of individuals are considered, it is known that the past experiences of individuals are often effective on their decisions. It can be argued that parents' experiences during this pandemic may influence their attitudes towards vaccination in the future years and potentially shape their decisions during the next pandemic as well (*Pertwee, Simas & Larson, 2022*). Supporting this information, the WHO recommends that the current pandemic should be examined from various aspects in order to be prepared for new health crises.

This study was conducted after the FDA approved the COVID-19 vaccine for use in children aged 5 to 11. The aim of the study was to assess parents' intentions regarding their children receiving the recommended vaccine during the pandemic period, as well as to investigate their attitudes towards the vaccine. This was an important time frame for determining parents' initial responses to a health authority's recommendation.

### Hypotheses of the study

Hypothesis 1: Parents' sociodemographic characteristics predict their Attitudes Towards COVID-19 Vaccine.

Hypothesis 2: Parents' Attitudes Towards COVID-19 Vaccine predict their intention to vaccinate their children against COVID-19.

Hypothesis 3: Parents' sociodemographic characteristics predict their intention to vaccinate their children against COVID-19.

Hypothesis 4: Parents' COVID-19 anxiety states predict their intention to vaccinate their children against COVID-19.

## MATERIALS & METHODS

### Study sampling

This cross-sectional study was conducted in 42 state primary schools located in the central region of Manisa, western Turkey. Within the scope of the school health program in Turkey, routine vaccination programs (childhood vaccination program) cover 13 different diseases such as measles, diphtheria, and tetanus during infancy and childhood. These vaccines are administered free of charge. However, at the time this study was conducted, the COVID-19 vaccine was not included in the routine vaccination program, and the decision on whether children would be vaccinated against COVID-19 had not yet been made by the Ministry of Health officials. The acceptance rate of vaccines within the scope of the routine vaccination program is 94% (*Gür, 2019*). Based on this information, it can be inferred that there is no widespread opposition to vaccination in the region where the research was conducted.

According to official records obtained from Manisa Provincial Directorate of National Education, the number of the parents with children aged 5–11 in the schools mentioned above is 19,530. With the Epi Info 2000 program, the minimum sample size required to achieve a 50% unknown prevalence and 5% margin of error, and 99% confidence level was calculated to be 642. In determining the number of participants to be reached from each school, the proportional sampling method was used according to the number of students in the schools.

As the students are young, many parents accompany their children to the school building every day. The data were collected from those parents who were present in the school building at the beginning or end of school hours, using the convenience sampling method. A total of 940 parents participated in the study ($n = 940$). Inclusion criteria for participation required proficiency in reading and writing Turkish, having a child aged between 5–11 years, and volunteering for the research. Data collection took place between January 1 and March 31, 2022, through face-to-face interviews (see Fig. 1). Each interview lasted between 10 to 15 min.

### Data collection and instruments

*Personal Information Form:* This form was prepared by the researchers according to the relevant literature and consisted of 23 questions on the parents' sociodemographic characteristics: age, gender, education level, income status, occupation, child's age, presence

With the data obtained from the Ministry of National Education, the number of primary schools in Manisa province was determined (62 primary schools in total).

With the data obtained from the Ministry of National Education, the number of primary schools in Manisa province was determined (62 primary schools in total).

Due to transportation difficulties, schools in rural areas were excluded (20 primary schools were excluded).

42 primary schools in Manisa city center were determined as the research center. The number of parents in these schools is 19530.

With the Epi Info 2000 program, the minimum sample size required to reach 50% unknown prevalence, 0.05 deviation and 99% confidence interval was calculated as 642.

The number of parents from each primary school was determined proportionally according to the number of students enrolled in the school.

Data was collected from 940 parents who met the research participation criteria from 42 primary schools between January 1 and March 31 (using convenience sampling method).

**Figure 1   Diagram of sampling method.**

of chronic illnesses, history of childhood vaccinations, personal history of COVID-19 infection, level of COVID-19 anxiety, sources of COVID-19 vaccine information, vaccination status, intention to vaccinate their children, and reasons for vaccine hesitancy (*Evren et al., 2022*; *Geniş et al., 2020*; *Szilagyi et al., 2021*).

*COVID-19 Anxiety Scale:* The scale was developed by *Evren et al. (2022)*. The participants reported the frequency of situations they experienced in the previous two weeks using a five-item Likert-type scale. The response options were: "Not at all", "Rarely, on less than one or two days", "A few days", "More than seven days", and "Almost every day in the last two weeks". The minimum score for each question was 0 and the maximum was 4. A total score ranging from 0 to 20 was calculated by adding the scores of each item. A higher score indicated greater anxiety in connection with COVID-19. Additionally, the scale has a cutoff point that creates a binary category. Scores of 5 and above from the scale indicate

anxiety, and low scores of 4 and below indicates no anxiety. The Cronbach alpha value in the study by *Evren et al. (2022)* was 0.80.

*Scale of Attitudes Towards COVID-19 Vaccine:* The scale was developed by *Geniş et al. (2020)*. The scale consists of nine items and two subscales. The first 4 items determine positive attitudes towards the COVID-19 vaccine. The last 5 items determine negative attitudes towards the COVID-19 vaccine. A high score from the positive attitude subscale indicates that positive attitudes towards the vaccine are high. A high score from the negative attitude subscale indicates that negative attitudes towards the vaccine are low. After the last 5 items in the negative attitudes subscale are reverse coded, the total score of the scale is obtained by summing all the items. The statements on the scale are evaluated as "I strongly disagree (1)", "I disagree (2)", "I am undecided (3)", "I agree (4)", and "I strongly agree (5)". The Cronbach alpha value in the study by *Geniş et al. (2020)* was 0.80.

The authors have permission to use both the "COVID-19 Anxiety Scale" and the "Scale of Attitudes Towards COVID-19 Vaccine" from the copyright holders.

## Statistical analysis

Data analysis was conducted using the program SPSS 25.0 (SPSS Inc., Chicago, IL, USA). In all analyses, the significance level was taken as $p < 0.05$. In the analysis of descriptive data, frequency, percentage, mean and standard deviation values were calculated. Student's $t$-test and One-Way ANOVA tests were utilized to compare the sociodemographic characteristics and COVID-19 anxiety status of the parents with their scores on the Scale of Attitudes Towards COVID-19 Vaccine. The Scale of Attitudes Towards COVID-19 Vaccine was assessed for normal distribution using skewness and kurtosis coefficients. Based on the values obtained by dividing the skewness and kurtosis coefficients by their respective standard errors, which fall between ±1.96, it can be concluded that the distribution of the Scale of Attitudes Towards COVID-19 Vaccine scores can be considered normal (*Can, 2019*; *Cevahir, 2020*). Therefore, the data was deemed suitable for normal distribution following the conducted procedures.

The COVID-19 Anxiety Scale was categorized into two groups: scores of 5 and above indicating anxiety, and scores of 4 and below indicating no anxiety (*Evren et al., 2022*). The average age of the parents was 38.82 years. The age variable was categorized into "38 years and below" and "39 years and above" based on the mean value.

To predict the variables influencing the parents' intentions to vaccinate their children against COVID-19, various characteristics were analyzed using a two-way logistic regression method. The parents' intentions were categorized into three initially: "Yes, I will accept"; "Undecided"; and "No, I will not accept". However, following literature recommendations (*Troiano & Nardi, 2021*), the "Undecided" and "No, I will not accept" responses were combined into a single category. Therefore, the intentions were reclassified into two categories for analysis: "Yes, I will accept"; and "No, I will not accept - I am undecided".

In the two-way comparisons, a logistical regression model was formed by including variables which created a significant correlation ($p < 0.05$). In the analysis, the Hosmer and Lemeshow test was employed to assess the Goodness of Fit. The model is considered appropriate if the $p$-value is $\geq \alpha$. Multicollinearity among independent variables in

logistic regression analysis was evaluated using correlation analysis, revealing no significant multicollinearity issues.

Additionally, Cronbach's alpha values were calculated for both the Scale of Attitudes Towards COVID-19 Vaccine and the COVID-19 Anxiety Scale to assess internal consistency reliability.

Effect size was determined by calculating Eta Square ($\eta$2) based on the average scores of the Scale of Attitudes Towards COVID-19 Vaccine among parent groups who indicated that they would "accept" the COVID-19 vaccine for their children *versus* those who were "undecided - will not accept" (*Serdar et al., 2021*).

The data collection form included an open-ended question. Those parents who did not want their children to be vaccinated against COVİD-19 were asked about their reasons. Qualitative answers given by some parents to this question were analyzed in the Nvivo 12 program (NVivo qualitative data analysis software; QSR International Pty Ltd. Version 12). The inductive analysis method was used. To increase the reliability of the qualitative analysis, two other researchers reviewed the raw data. The emerging theme was largely agreed upon. During the qualitative analysis process, the thematic analysis steps recommended by Braun et al. were followed (*Braun et al. 2019*).

### Ethical consideration

Before starting the research, approval was obtained from the Manisa Celal Bayar University Health Sciences Ethics Committee (February 2022, Issue: 20.478.486/1192). Volunteer parents were informed about the study, and their verbal and written consent was obtained. In every process of the research, the Declaration of Helsinki was followed.

## RESULTS

In this study with 940 participants, the effect size was evaluated with an Eta square value of 0.163 ($p < 0.05$). The Cronbach's alpha values for the Scale of Attitudes Towards COVID-19 Vaccine and the COVID-19 Anxiety Scale were found to be 0.89 and 0.79, respectively.

In the study, a demographic profile of the participants revealed that 51.6% of parents were aged 39 years or older, with a majority (63.2%) being female. Nearly half of the participants (49.0%) had attained a university degree, and a significant proportion (28.7%) were employed in government positions. A majority (64.4%) reported having income levels that matched their expenditures (poverty limit for Türkiye). Furthermore, a substantial portion of parents indicated that their children had no chronic illnesses (90.5%) and had received childhood vaccinations (90.5%) (Table 1).

In the research, the findings revealed that 51.3% of the participants reported that at least one family member had contracted COVID-19. A significant majority of the parents (93.7%) did not express anxiety regarding COVID-19, while 69.8% received information about COVID-19 vaccination. Notably, 48.7% obtained COVID-19 information from press and media sources, and 85.1% had received a COVID-19 vaccination (Table 2). Regarding the parents' intentions regarding COVID-19 vaccination for their children, 50.3% indicated they would not accept it. Among those who expressed vaccine hesitancy,

**Table 1   Distribution of descriptive characteristics.**

| Variables | | n | % |
|---|---|---|---|
| **Age**  (38.82 ± 5.36, Min:25, Maks:61) | 38 years and under | 455 | 48.4 |
| | 39 years and more | 485 | 51.6 |
| **Gender** | Female | 600 | 63.8 |
| | Male | 340 | 36.2 |
| **Education Level** | Primary education | 196 | 20.9 |
| | High school | 283 | 30.1 |
| | University | 461 | 49.0 |
| **Employment status** | Not working | 262 | 27.9 |
| | Employee | 260 | 27.7 |
| | Government employees | 270 | 28.7 |
| | Self-employment | 108 | 11.5 |
| | Retired | 40 | 4.2 |
| **Income status** | Income less than expenses | 180 | 19.4 |
| | Income equals expense | 605 | 64.4 |
| | Income more than expenses | 152 | 16.2 |
| **Child's age**  (9,26 ± 1,99, Min:6, Maks:13) | 9 years and under | 488 | 51.9 |
| | 10 years and more | 452 | 48.1 |
| **Childhood chronic illness** | Yes | 81 | 8.6 |
| | No | 859 | 91.4 |
| **Childhood vaccination status** | Yes | 851 | 90.5 |
| | No | 89 | 9.5 |

53.3% cited the need for detailed investigation into the vaccine's effects on children, 17.0% expressed concerns about potential harm to their children from the vaccine, and 12.4% mentioned that their child had recovered from COVID-19, thus not seeing a need for vaccination. Additionally, 11.6% believed that the vaccine would be ineffective, and 5.8% cited their child's existing chronic illness as a reason against vaccination (Table 3).

When the sociodemographic characteristics of the parents and the mean scores of the COVID-19 Vaccination Attitude Scale were examined, it was determined that there was a significant relationship between the scale scores and the variables of age, occupation, income level, parents' intentions to vaccinate their children against COVID-19, acceptance status of childhood vaccinations, receiving information about the COVID-19 vaccine, and parents' COVID-19 anxiety status ($p < 0.05$). No significant difference was found when the other variables and the total and subdimension mean scores on the Attitudes to COVID-19 Vaccination Scale were examined ($p > 0.05$) (Table 4). These results form the basis of hypotheses 1–2.

A logistic regression model was set up, including variables which created a significant correlation ($p < 0.05$) in the two-way comparisons. In the logistic regression analysis, several significant factors were identified influencing the parents' intentions regarding COVID-19 vaccination for their children. Those parents aged 39 or older were 1.522 times more likely to express intention to accept vaccination (''yes I will accept''). Those who themselves had received the COVID-19 vaccine were 3.785 times more likely to intend

**Table 2 Distribution of data on parental COVID-19 and COVID-19 vaccine.**

| Variables | | n | % |
| --- | --- | --- | --- |
| **Family history of COVID-19 disease** | Yes | 482 | 51.3 |
| | No | 458 | 48.7 |
| **COVID-19 anxiety status of parents** | Yes 0 $\leq$4 | 881 | 93.7 |
| | No 5 $\geq$10 | 59 | 6.3 |
| **Status of getting information about the COVID-19 vaccine** | Yes | 656 | 284 |
| | No | 69.8 | 30.2 |
| **Who received information about the COVID-19 vaccine?** | Doctor | 244 | 31.0 |
| | Midwife and nurse | 69 | 8.8 |
| | Press and publishing organs | 384 | 48.7 |
| | Relatives and neighbors | 29 | 3.7 |
| | the other | 62 | 7.8 |
| **Parents' COVID-19 vaccination status** | Vaccinated | 800 | 85.1 |
| | Unvaccinated | 140 | 14.9 |
| **Total** | | 940 | 100 |

**Table 3 Information on parents' intentions to vaccinate their children against COVID-19.**

| Variables | | n | % |
| --- | --- | --- | --- |
| **Parents' intentions to vaccinate their children against COVID-19** | Yes I will accept (a) | 188 | 20.0 |
| | No, I will not accept (b) | 473 | 50.3 |
| | I am undecided(c) | 279 | 29.7 |
| **Total** | | 940 | 100 |
| **Reasons for parents who do not want their children to be vaccinated against COVID-19\*** | They thought that more advanced research was needed on the effect of the vaccination on children | 396 | 53.3 |
| | They thought that the vaccine would be harmful to their children | 128 | 17.0 |
| | Their child had recovered from COVID-19. They stated that their child would not need to be vaccinated against COVID-19 | 93 | 12.4 |
| | They thought that the vaccine would be useless | 87 | 11.5 |
| | They stated that would not vaccinate their child because they had a chronic disease | 47 | 5.8 |
| **Total** (\*752 participants answered this question) | | 752 | 100 |

to vaccinate their children. Similarly, the parents whose children had received childhood vaccinations were 2.776 times more likely to express intention to vaccinate their children for COVID-19. Furthermore, those who had received information about the COVID-19 vaccination program were 2.110 times more likely to intend to vaccinate their children. Lastly, the parents who reported anxiety about contracting COVID-19 were 3.918 times more likely to express intention to vaccinate their children ("yes I will accept") (Table 5). These findings support hypotheses 3–4

**Table 4** Evaluation of socio-demographical variables and mean scores of the attitudes towards COVID-19 vaccine scale.

| Variables | n | % | Positive attitudes Ort ± SS | Negative attitudes Ort ± SS | Total Ort ± SS |
|---|---|---|---|---|---|
| **Age** | | | | | |
| 38 years and under | 455 | 48.4 | 13.37 ± 4.26 | 16.88 ± 4.00 | 30.25 ± 7.49 |
| 39 years and more | 485 | 51.6 | 14.01 ± 4.11 | 17.63 ± 4.07 | 31.64 ± 7.35 |
| t(p) | | | 2.33 (0.02) | 2.85 (0.00) | 2.87 (0.00) |
| **Gender** | | | | | |
| Female | 600 | 63.8 | 13.70 ± 4.10 | 17.14 ± 3.80 | 30.84 ± 7.15 |
| Male | 340 | 36.2 | 13.71 ± 4.36 | 17.50 ± 4.45 | 31.20 ± 7.96 |
| t(p) | | | 0.018 (0.98) | 1.33 (0.18) | 0.71 (0.47) |
| **Education Level** | | | | | |
| Primary education | 196 | 20.9 | 13.35 ± 3.94 | 17.09 ± 3.84 | 30.44 ± 6.63 |
| High school | 283 | 30.1 | 14.03 ± 4.12 | 17.16 ± 4.29 | 31.20 ± 7.65 |
| University | 461 | 49.0 | 13.70 ± 4..34 | 17.41 ± 3.99 | 31.06 ± 7.63 |
| F(p) | | | 1.6133 (0.20) | 0.573 (0.56) | 0.662 (0.51) |
| **Employment status** | | | | | |
| Not working (a) | 262 | 27.9 | 13.27 ± 4.10 | 16.83 ± 3.91 | 29.95 ± 7.11 |
| Employee (b) | 260 | 27.7 | 13.95 ± 4.07 | 17.54 ± 4.18 | 31.29 ± 7.47 |
| Government employees (c) | 270 | 28.7 | 14.00 ± 4.20 | 17.77 ± 3.88 | 31.77 ± 7.45 |
| Self-employment (d) | 108 | 11.5 | 13.23 ± 4.30 | 16.75 ± 4.41 | 29.89 ± 7.84 |
| Retired (e) | 40 | 4.2 | 14.90 ± 4.85 | 18.27 ± 4.11 | 33.17 ± 7.92 |
| F(p) | | | 2.50 (0.04) | 2.97 (0.01) | 3.27 (0.01) |
| Post Hoc test-Tukey | | | a=d<b=c<e | a=d<b=c<e | a=d<b=c<e |
| **Income status** | | | | | |
| Income less than expenses (a) | 180 | 19.4 | 12.93 ± 4.36 | 16.42 ± 4.20 | 29.35 ± 7.79 |
| Income equals expense (b) | 605 | 64.4 | 13.85 ± 4.08 | 17.39 ± 3.93 | 31.24 ± 7.19 |
| Income more than expenses (c) | 152 | 16.2 | 14.15 ± 4.37 | 17.88 ± 4.23 | 31,91 ± 7.85 |
| F(p) | | | 3,89 (0.02) | 5.52 (0.00) | 5.69 (0.00) |
| Post Hoc test-Tukey | | | a<b<c | a<b<c | a<b<c |
| **Child's age** | | | | | |
| 9 years and under | 488 | 51.9 | 13.58 ± 4.29 | 17.11 ± 3.91 | 30.69 ± 7.52 |
| 10 years and more | 452 | 48.1 | 13.83 ± 4.09 | 17.44 ± 4.13 | 31.27 ± 7.35 |
| t(p) | | | 0.920 (0.35) | 1.238 (0.21) | 1.192 (0.32) |
| **Family history of COVID-19 disease** | | | | | |
| Yes | 458 | 48.7 | 13.83 ± 4.22 | 17.29 ± 4.07 | 31.12 ± 7.59 |
| No | 482 | 51.3 | 13.68 ± 4.13 | 17.27 ± 3.98 | 30.95 ± 7.33 |
| F(p) | | | 1.50 (0.19) | 1.96 (0.09) | 1.86 (0.11) |
| **Parents' intentions to vaccinate their children against COVID-19** | | | | | |
| Yes I will accept (a) | 188 | 20.0 | 17.06 ± 3.52 | 19.95 ± 3.88 | 37.02 ± 6.55 |
| No, I will not accept (b) | 473 | 50.3 | 11.88 ± 3.95 | 15.90 ± 3.80 | 27.79 ± 6.82 |
| I am undecided(c) | 279 | 29.7 | 14.51 ± 3.26 | 17.78 ± 3.56 | 32.30 ± 599 |
| F(p) | | | 143.64 (0.00) | 82.14 (0.00) | 142.36 (0.00) |
| Post Hoc test-Tukey | | | b<c<a | b<c<a | b<c<a |

**Table 4** (*continued*)

| Variables | n | % | Positive attitudes Ort ± SS | Negative attitudes Ort ± SS | Total Ort ± SS |
|---|---|---|---|---|---|
| **Parents' COVID-19 vaccination status** | | | | | |
| Vaccinated | 800 | 85.1 | 14.29 ± 3.88 | 17.62 ± 3.92 | 31.91 ± 7.03 |
| Unvaccinated | 140 | 14.9 | 10.33 ± 4.34 | 15.27 ± 4.21 | 25.61 ± 7.55 |
| *t(p)* | | | 10.91 (0.00) | 6.43 (0.00) | 9.66 (0.00) |
| **Childhood vaccination status** (*Covid-19 vaccine not included*) | | | | | |
| Yes | 851 | 90.5 | 13.80 ± 4.20 | 1727 ± 4.06 | 31.08 ± 7.50 |
| No | 89 | 9.5 | 1314 ± 4.11 | 17.24 ± 4.03 | 30.39 ± 7.18 |
| *t(p)* | | | 1.76 (0.07) | 0.07 (0.93) | 1.03 (0.30) |
| **Status of recive information about the COVID-19 vaccine** | | | | | |
| Yes | 656 | 69.8 | 13.98 ± 4.24 | 17.47 ± 4.15 | 31.45 ± 7.62 |
| No | 284 | 30.2 | 13.05 ± 4.02 | 16.81 ± 3.78 | 29.86 ± 6.93 |
| *t(p)* | | | 3.12 (0.00) | 2.28 (0.02) | 3.00 (0.00) |
| **COVID-19 anxiety status of parents (According to the scale score)** | | | | | |
| Yes | 881 | 93.7 | 13.50 ± 4.16 | 17.04 ± 3.96 | 30.54 ± 7.29 |
| No | 59 | 6.3 | 16.71 ± 3.43 | 20.66 ± 3.95 | 37.37 ± 6.87 |
| *t(p)* | | | 5.78 (0.00) | 6.78 (0.00) | 6.98 (0.00) |

Notes.
t. Student t Testi; F, One-way ANOVA.

**Table 5   Predictors of parents' intentions to vaccinate their children with COVID-19.**

| Variables | Category | Odds ratio | 95% CI For OR | | *p* |
|---|---|---|---|---|---|
| | | | Lower | Upper | |
| **Age** | 38 years and under | 1 | | | 0.01 |
| | 39 years and more | 1.522 | 1.110 | 2.169 | |
| **Parents' COVID-19 vaccination status** | No | 1 | | | <0.001 |
| | Yes | 3.785 | 1,861 | 7,698 | |
| **Childhood vaccination status** | No | 1 | | | 0.01 |
| | Yes | 2.776 | 1.278 | 6.030 | |
| **Status of recive information about the COVID-19 vaccine** | No | 1 | | | <0.001 |
| | Yes | 2.110 | 1.396 | 3.191 | |
| **COVID-19 anxiety status of parents** | No | 1 | | | <0.001 |
| | Yes | 3.918 | 2.204 | 6.963 | |

Notes.
Model Hosmer–Lemeshow test $X^2(7)$ = 7,636, p = 0.36.
Cox and Snell $R^2$ = 0.081; Nagelkerke $R^2$ = 0.12.

282 of the parents who stated that they would not vaccinate their children against COVID-19 or were undecided wrote the reason for their thoughts on the data collection form. These were short statements of a few sentences. As a result of the inductive analysis method, the theme of "distrust" emerged. One parent's comment about the process was: I don't want to open Pandora's box. I am concerned about the future effects of the vaccine (NG; age; 37, Gender; female).

## DISCUSSION

This study was conducted shortly after the FDA publicly recommended vaccinating children against COVID-19. Around the same time, the vaccination of children aged 5–11 against COVID-19 became a prominent topic of discussion in the Turkish public sphere. Social media and television programs featured two prevailing opinions: one group arguing that COVID-19's impact on children was minimal and that vaccination was therefore unnecessary, while another group advocated for vaccinating children to prevent infection. Amidst these debates during the pandemic, it was crucial to investigate parents' initial responses to health authority recommendations regarding COVID-19 vaccination for their children.

The study found that under the above-mentioned conditions, only a fifth (20%) of the parents were willing to vaccinate their children against COVID-19. In fact, it was surprising that the parents had little intention of vaccinating their children against COVID-19 because in the region where the research was conducted, the acceptance rate of childhood vaccines was over 90% (*Gür, 2019*). It can be stated that these results indicate a special situation for the COVID-19 vaccine. It may have resulted from the communication policies of health authorities or government officials during the pandemic period. They may not have been able to explain the COVID-19 vaccination program to the public well (*Laebens & Öztürk, 2022*).

Similarly, studies conducted in various countries have shown varying levels of parental intention to vaccinate their children against COVID-19. In Jordan, only 30.2% of parents planned to vaccinate their children against COVID-19 (*Al-Qerem et al., 2022*). Meanwhile, a study conducted in England in 2020 indicated that 48.4% of parents intended to vaccinate their children against COVID-19 (*Bell et al., 2020*). Before the FDA's recommendation, a study conducted in Turkey found that 38% of parents planned to have their children vaccinated against COVID-19 (*Yılmaz & Sahin, 2021*).

When comparing the findings of previous studies to those of this study, it becomes apparent that the parents' intentions to vaccinate their children against COVID-19 were relatively lower than in the early stages of the pandemic. This decrease is noteworthy and could stem from several potential reasons. Two main possibilities emerge as explanations for this trend. Firstly, it is possible that the parents' perceived threat of COVID-19 diminished over time, leading to reduced urgency in vaccinating their children. Secondly, the decline in the parents' confidence and positive attitudes towards the COVID-19 vaccine may also contribute to this shift in intention.

However, it is anticipated that the endorsement of a COVID-19 vaccine by health authorities would alleviate both of these conditions. When a vaccine is recommended by a health authority, it conveys a message to parents that the threat of COVID-19 continues to pose risks. Furthermore, it is hoped that such recommendations would reassure parents about the vaccine's safety and efficacy. However, according to the findings of this study, these expectations do not appear to have materialized fully. At the time of the research, there seems to have been erosion in parents' trust towards health authorities, as indicated by their attitudes towards COVID-19 vaccination. The qualitative data from the study reinforced

these findings, revealing widespread lack of confidence in the COVID-19 vaccine among the parents. Moreover, as shown in Table 3, primary concerns expressed by the participating parents who were hesitant to vaccinate their children included the need for further research into the vaccine's effects on children, concerns about potential harm from the vaccine, and doubts about its effectiveness. These findings align with similar concerns raised in studies conducted in other countries, such as those by *Szilagyi et al. (2021)* in the USA.

The findings of the study suggest that the parents demanded more reliable and transparent information to build trust in COVID-19 vaccines. *Opel et al. (2023)* recommend that sharing vaccine study results in a clear and understandable manner with parents could help enhance trust. It can be said that mainstream media, health experts and government officials need to take a more active role in this regard.

When evaluating the factors influencing the participating parents' attitudes towards the COVID-19 vaccine and their intention to vaccinate their children, it was observed that the parents aged 39 and older exhibited more positive attitudes towards the vaccine compared to the younger parents. Specifically, these older parents were 1.522 times more likely to intend to vaccinate their children against COVID-19. In the literature, it is noted that as individuals age, their resilience against environmental factors decreases (*Detoc et al., 2020*; *Zdziarski et al., 2021*). Given that COVID-19 tends to be more severe in older individuals, it can be inferred that the parents' heightened risk perception positively influenced their attitude towards vaccination, potentially motivated by a desire to protect both themselves and their children. These findings are consistent with previous studies highlighting age-related differences in vaccine acceptance (*Davis et al., 2020*; *Temsah et al., 2021*).

When the socio-economic status of the parents, their COVID-19 vaccination attitude and their intention to have their children vaccinated against the COVID-19 were evaluated, it was seen that the vaccination attitudes of the parents with a high income were more positive. In a different study, similar to these results, it was determined that increasing income status positively affected attitudes towards the COVID-19 vaccine (*Fisher et al., 2020*). However, it was observed that income status did not affect parents' intention to vaccinate their children.

In the study, it was observed that the variables parental gender, educational status, and number of children did not affect attitudes towards the COVID-19 vaccine. Similar to this result, another study conducted in Turkey found that the gender variable and education level did not have a significant effect on parents' vaccination attitudes (*Başal & Emir Öksüz, 2022*). On the other hand, it was found that the COVID-19 vaccine attitudes of parents who planned to have their children vaccinated against COVID-19 were significantly positive. It can be inferred that this result is expected.

The attitudes of the parents who received the COVID-19 vaccine were positive towards the COVID-19 vaccine. At the same time, it was also determined that these parents were 3.785 times more likely to have their children vaccinated against COVID-19. Supporting this result, a study conducted in the USA in 2021 revealed that parents who were vaccinated against COVID-19 were more willing to vaccinate their children against COVID-19 (*Szilagyi et al., 2021*; *Temsah et al., 2021*). It can be stated that these parents' experiences

with the COVID-19 vaccine positively influenced their decision about this vaccine recommended for their children.

The acceptance rate of childhood vaccines (excluding COVID-19 vaccines) among children whose parents participated in the study was 90.5%. It was found that the parents who accepted childhood vaccinations for their children showed more positive attitudes towards COVID-19 vaccination compared to those who did not. Specifically, these parents were 2.776 times more likely than others to accept the recommended COVID-19 vaccine for their child. This finding suggests that parents' past behaviors related to childhood vaccinations influence their attitudes towards newly recommended vaccines, such as the COVID-19 vaccine.

It was found that the parents who received information about the COVID-19 vaccine had more positive attitudes towards the COVID-19 vaccine than those who did not. Additionally, It was also revealed that these parents who received information were 2.110 times more likely to have their children vaccinated against COVID-19. Similarly, in a study conducted in Germany, it was determined that parents who received information about the COVID-19 pandemic and vaccination were more likely to have their children vaccinated against COVID-19 if a vaccine was recommended for their children (*Brandstetter et al., 2021*; *Zhang et al., 2020*; *Al-Qerem et al., 2022*). This indicates that reliable information flow positively affects attitudes towards vaccines, as suggested by *Opel et al. (2023)*.

Anxiety, fear and individual risk perception seem to be important determinants of vaccine acceptance (*Bendau et al., 2021*). In this research, it was determined that those parents with high COVID-19 anxiety had positive attitudes towards COVID-19 vaccination, as expected. Similarly, studies conducted in the United Kingdom (*Salali & Uysal, 2022*), France (*Detoc et al., 2020*) and the United States (*Head et al., 2020*; *Nguyen et al., 2021*) found that people with high COVID-19 anxiety levels had positive attitudes towards the COVID-19 vaccine. Additionally, it was found that parents with high COVID-19 anxiety were 3.918 times more likely to vaccinate their children against COVID-19.

## LIMITATIONS

This research has several limitations that should be acknowledged. Firstly, the study was conducted exclusively in the city center of Manisa, situated in western Turkey, which may limit the generalizability of the findings to broader populations. Additionally, the reasons provided by the parents for their reluctance to vaccinate their children against COVID-19 were derived solely from brief statements written on the scales. A more comprehensive understanding could have been achieved through a mixed-method approach combining qualitative and quantitative methodologies, allowing for deeper exploration and interpretation of parental perspectives.

## CONCLUSIONS

The findings highlight that a significant majority of the parents expressed hesitancy towards a COVID-19 vaccine recommended for children by health authorities, despite their own vaccination coverage against COVID-19. Sociodemographic factors and levels

of COVID-19 anxiety among the participating parents were identified as predictors influencing their attitudes towards the COVID-19 vaccine and their intention to vaccinate their children. Qualitative insights underscored challenges in governmental and health authority communication regarding vaccination programs and public health practices during the pandemic. Confidence emerged as a critical factor in decisions about protecting children through vaccination. These results serve as a timely warning signal, emphasizing the need for proactive strategies in future pandemic preparedness efforts.

## ACKNOWLEDGEMENTS

We thank the parents who participated in the research.

### Funding
The authors received no funding for this work.

### Competing Interests
The authors declare there are no competing interests.

### Author Contributions
- Nurcan Çelik Odabaşı conceived and designed the experiments, performed the experiments, authored or reviewed drafts of the article, and approved the final draft.
- Ali Tayhan conceived and designed the experiments, analyzed the data, prepared figures and/or tables, authored or reviewed drafts of the article, and approved the final draft.
- Gulengul Mermer conceived and designed the experiments, prepared figures and/or tables, authored or reviewed drafts of the article, and approved the final draft.

### Human Ethics
The following information was supplied relating to ethical approvals (i.e., approving body and any reference numbers):
Celal Bayar Üniversity Health Sciences Ethics Committee

### Data Availability
The raw data is available in the Supplemental File.

### Supplemental Information
Supplemental information for this article can be found online at http://dx.doi.org/10.7717/peerj.18056#supplemental-information.

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
