# Peer review of "Attitudes and intentions of parents towards the COVID-19 vaccine for their children at a special moment of the pandemic"

_PeerJ, doi:10.7717/peerj.18056_

## Round 0.1 · original submission · Major Revisions

The paper addresses a relevant and current topic, considering the importance of understanding parents' attitudes towards COVID-19 vaccination for children. However, there are significant areas that need improvement to enhance the robustness and impact of the study. The introduction lacks a more in-depth presentation of the state of the art; to clearly position the relevance of the current study, it is crucial to incorporate a more comprehensive review of the existing literature, including a detailed discussion on the factors influencing vaccine hesitancy, both in general contexts and specific to the COVID-19 pandemic, as well as comparative international studies. The discussion should be more closely aligned with the extensive literature available on the subject, comparing and contrasting the results with findings from other relevant studies, highlighting similarities and differences, and addressing the implications for public policies and health communication strategies. The statistical analysis conducted is adequate, but there are opportunities for improvement that could increase the study's impact, such as the use of additional statistical techniques (see, for example, multilevel analysis), including sensitivity analyses or interaction models, to provide a more robust and detailed understanding of the data, and the analysis of interactions between different demographic and psychosocial variables, which could reveal deeper insights into the factors influencing vaccine hesitancy. Despite the identified limitations, this study provides a valuable contribution to understanding parents' attitudes towards COVID-19 vaccination for children. The suggested improvements, particularly in the introduction, discussion, and statistical analysis, can significantly increase the robustness and impact of the study, providing a stronger basis for public health interventions and strategic communication. With these improvements, the study could offer more robust and impactful insights into parents' attitudes towards COVID-19 vaccination for children.

Reviewer 1 ·

Basic reporting

The manuscript presents the results of observational research on parents' attitudes and intentions about vaccinating their children. The text can be better organized to facilitate reading and understanding of the results. At some points, the text is vague (for details see attachment)
The research has the potential to provide interesting information about the factors that can influence parents' decisions to vaccinate their children and follow recommendations from health authorities. The results can help in the development of risk communication strategies. However, the text still needs improvement and several issues surrounding the analyzes must be clarified to judge the validity of the results.

Experimental design

The study is observational and used different instruments that were answered by the participants. Lack of description of the studied population (parents, location, sociodemographic data of the country/region, public health policies (including vaccination programs)). Some variables were not described, nor the purpose of the static tests based on the research objectives (especially whether the assumptions were met).

Validity of the findings

It is difficult to assess the validity of the results due to the lack of clarity regarding some variables, statistical tests and compliance with their respective requirements.

Additional comments

no comment

Annotated reviews are not available for download in order to protect the identity of reviewers who chose to remain anonymous.

Reviewer 2 ·

Basic reporting

This study investigates parental intentions regarding the COVID-19 vaccine for children aged 5 to 11 following FDA approval. The research aimed to assess parents' initial responses to health authority recommendations during this crucial period in the pandemic. Results revealed that while 85.1% of parents had received the COVID-19 vaccine themselves, only 20% intended to vaccinate their children. Parental intentions were influenced by factors such as parent age, COVID-19 anxiety level, parental vaccination status, and knowledge about the COVID-19 vaccine. Despite FDA approval, the study highlights significant parental hesitancy regarding COVID-19 vaccination for children, with more than two-thirds expressing reluctance. These findings underscore the importance of addressing parental concerns and providing targeted education to increase vaccine acceptance in pediatric populations. Nonetheless, I would like to suggest some revisions to enhance the quality of your work.

1. Introduction: Could you please provide some background on the timeline of vaccine administration in the study region? Information on when vaccines were first administered to adults and children, specifically for the age group of 5-11, would be valuable. Additionally, could you include the vaccination rate among children aged 5-11 during the study period, based on a representative sample?

Experimental design

1. Methodology (Geographic Specificity): In lines 78-79, it would be beneficial to specify the geographic region where the study was conducted to better contextualize the findings.

2. Methodology (Participant Selection): Please outline the eligibility criteria for participant selection in the study sampling section. Consider including whether understanding a certain language was a criterion and if schools within specific geographic ranges were targeted. A flow chart illustrating how the final sample size was determined might also enhance clarity for readers.

3. Measurement Clarification: The methodology for measuring the receipt of information, vaccination status, and the scales for COVID-19 anxiety and attitudes towards vaccination remains unclear. Could you specify which vaccinations were considered? Additionally, including the questionnaire as a supplementary file would aid in understanding these measures.

4. Data Analysis (COVID-19 Anxiety): Please clarify how the binary variable for COVID-19 anxiety was derived in Table 2. The threshold for classification into 'anxious' versus 'not anxious' categories based on scale values would benefit from further explanation.

5. Statistical Methods (Cronbach's Alpha): The Cronbach alpha values for the scales used in the study are mentioned in lines 104 and 112. It would be appropriate to discuss these statistical methods in the “Statistical Analysis” section and present the results in the “Results” section.

6. Clarification of Logistic Regression Model: In lines 120-124, you mention using a "two-way logistic regression model." Since this term is non-standard, could you specify if this refers to a binary logistic regression with combined response categories or a multinomial logistic regression with three outcome levels: yes, no, and hesitant? It would also be helpful to define this earlier in the methodology and clarify which outcome's odds ratios are presented in Table 5.

Validity of the findings

1. Results (Participant Numbers): At the beginning of the results section (lines 133-135), please state the total number of participants included and refer back to the eligibility flow chart. This helps in establishing the context of the findings.

2. Organization of Qualitative Findings: The qualitative findings presented between lines 197 and 203 might be more appropriately discussed within the method and results sections, ensuring a logical flow and coherence in reporting.

3. Discussion (Sample Representativeness): In the first paragraph of the Discussion, it would be insightful to address whether the sample can be considered representative of the region. The low willingness to vaccinate among parents, even post-FDA approval, raises concerns about the generalizability and impact of your findings.

---

## Round 0.2 · Minor Revisions

After a careful review of the revised version of the manuscript, I confirm that the authors have addressed majority of the reviewers' comments. The revision was thorough, and the points raised were appropriately addressed. But, the reviewer #1 have additional comments.

Reviewer 1 ·

Basic reporting

The manuscript has shown significant improvements. The authors have answered all questions and included the requested information.

The introduction is still timid and cites studies in the United Kingdom and Italy, while a study conducted with the Turkish population appears only in the discussion. This survey should be in the introduction, since it refers precisely to the country in question.
I also recommend that there be more detail on the pandemic context in Turkey (first case, health policies, death toll, anti-vaccine propaganda)

The introduction describes hypotheses, but these are described very broadly. For example, sociodemographic characteristics predict attitudes and intention to vaccinate children. Hypotheses must be accurate (for example, low income predicts X). Sociodemographic characteristics are a set of factors, often correlated. The text states in hypothesis 1 that sociodemography predicts attitude (a-->b), in hypothesis 2 that attitude predicts intention (b-->c), so one would expect that "a" predicts "c" (hypothesis 3). If this was the study design, then a mediational analysis is necessary. In fact, the moderating role of attitude on behavior is a well-known interaction in the literature.

Experimental design

The Materials and Methods section has improved considerably. I would suggest that the schools be described better (are they public? Are they in more or less vulnerable neighborhoods) and the source that allowed us to calculate the total number of children aged 5-11.

The excerpt "The data were collected from those parents who were present in the school building at the beginning and end of school hours" suggests that the respondents had to be at the beginning and end of school hours... so there are two conditions that needed to be met. If it is only one or the other, then correct it to "or".

In the excerpt "Additionally, the scale has a cutoff point that creates a binary category. Scores of 5 and above from the scale indicate anxiety, and low scores of 4 and below indicate no anxiety." Assuming that the instrument consisted of 5 items, each with points ranging from 0-4, would someone who answered 1 for all items (rarely) have anxiety? And would that person have the same response compared to someone with a score of 20? Note that the study deliberately lost the variability of the data by choosing to transform a continuous/ordinal data into a categorical one. Can you explain the reason for this decision?

Validity of the findings

The description of the results is also better, but again some points raise doubts. For example, "a majority (64.4%) reported having income levels that matched their expenditures". Is someone with an income equal to 100 dollars in the same category as someone who has an income equal to 1,000,000 dollars as long as both are able to match their respective incomes with their financial obligations? Please explain better.

The discussion (as well as the introduction) could be expanded. For example, the political context of Turkey itself (e.g. https://www.cambridge.org/core/journals/government-and-opposition/article/abs/erdogan-governments-response-to-the-covid19-pandemic-performance-and-actuality-in-an-authoritarian-context/8A254B11781A1C547BAF38513B4E8DC1). It was also not clear whether the vaccine for children (and the general public) was provided by the government's health program (the text says that it is not included in the childhood vaccination program). If the population needed to pay for the vaccine, then it could be a factor in supporting childhood vaccination, but not necessarily vaccinating against covid. Government propaganda may also have affected the population's perception of risk... anyway, the introduction and discussion could be expanded further.

Additional comments

none

Reviewer 2 ·

Basic reporting

The authors have made comprehensive revisions to address all previous comments. They provided detailed background information on the vaccine administration timeline and vaccination rates for children aged 5-11. The methodology section now includes geographic specificity, clear participant selection criteria, and a flow chart for sample size determination. Measurement methods, data analysis procedures, and statistical methods are clarified, including the explanation of COVID-19 anxiety classification and the logistic regression model. The revised manuscript improves the organization of qualitative findings and discusses the sample's representativeness, enhancing the overall clarity and robustness of the study.

Experimental design

NA

Validity of the findings

NA

---

## Round 0.3 · accepted · Accept

After a careful review of the revised version of the manuscript, I confirm that the authors have addressed the reviewer comments. The revision was thorough, and the points raised were appropriately addressed